# The PI3K-Akt-mTOR Signaling Pathway in Human Acute Myeloid Leukemia (AML) Cells

**DOI:** 10.3390/ijms21082907

**Published:** 2020-04-21

**Authors:** Ina Nepstad, Kimberley Joanne Hatfield, Ida Sofie Grønningsæter, Håkon Reikvam

**Affiliations:** 1Department of Clinical Science, University of Bergen, N-5021 Bergen, Norway; Ina.Nepstad@nsd.no (I.N.); Kimberley.Hatfield@uib.no (K.J.H.); Ida.Gronningseter@uib.no (I.S.G.); 2Department of Immunology and transfusion medicine, Haukeland University Hospital, N-5021 Bergen, Norway; 3Department of Medicine, Haukeland University Hospital, N-5021 Bergen, Norway

**Keywords:** Akt, AML, chemotherapy, metabolism, mTOR, PI3K, signaling

## Abstract

Acute myeloid leukemia (AML) is a heterogeneous group of diseases characterized by uncontrolled proliferation of hematopoietic stem cells in the bone marrow. Malignant cell growth is characterized by disruption of normal intracellular signaling, caused by mutations or aberrant external signaling. The phosphoinositide 3-kinase (PI3K)-Akt-mammalian target of rapamycin (mTOR) pathway (PI3K-Akt-mTOR pathway) is among one of the intracellular pathways aberrantly upregulated in cancers including AML. Activation of this pathway seems important in leukemogenesis, and given the central role of this pathway in metabolism, the bioenergetics of AML cells may depend on downstream signaling within this pathway. Furthermore, observations suggest that constitutive activation of the PI3K-Akt-mTOR pathway differs between patients, and that increased activity within this pathway is an adverse prognostic parameter in AML. Pharmacological targeting of the PI3K-Akt-mTOR pathway with specific inhibitors results in suppression of leukemic cell growth. However, AML patients seem to differ regarding their susceptibility to various small-molecule inhibitors, reflecting biological heterogeneity in the intracellular signaling status. These findings should be further investigated in both preclinical and clinical settings, along with the potential use of this pathway as a prognostic biomarker, both in patients receiving intensive curative AML treatment and in elderly/unfit receiving AML-stabilizing treatment.

## 1. Introduction

### 1.1. Acute Myeloid Leukemia

Acute myeloid leukemia (AML) is an aggressive malignancy characterized by rapid proliferation of immature myeloid leukemia cells [1,2]. In most cases, this disease involves a bone marrow malignancy, although malignant cells may also be detected in peripheral blood or as extramedullary infiltration [1,3,4]. The acute promyelocytic leukemia (APL) variant of AML is characterized by specific genetic abnormalities in the leukemic cells and severe coagulopathy, and treatment differs from that of other types of AML [5,6,7]. In this review, we use the term AML to refer to non-APL variants of the disease.

For a significant majority of patients, AML primarily affects the bone marrow, and by definition, at least 20% of the nucleated bone marrow cells are immature or undifferentiated leukemia blasts [2]. Based on etiology, AML could be classified as de novo, secondary, i.e., after previously other hematological malignancies or after chemoradiotherapy for other malignant diseases, or relapsed/refractory, after reoccurrence of the disease after treatment or lack of treatment response [8]. Cytogenetic and molecular genetic aberrations in AML have additional diagnostic and prognostic impacts [1,9]. Several strategies for subclassification, such as micro-RNA, epigenetic, transcriptomic, metabolomic, and proteomic characterization, have been suggested, although are currently not a part of routine handling of these patients [10,11,12,13,14,15]. Intensive chemotherapy is the only treatment that can cure AML, and autologous or allogeneic stem cell transplantation can be a part of this intensive treatment [3]. For patients who cannot tolerate intensive treatment, e.g., elderly patients over 70–80 years of age, and patients with severe comorbidity, the most commonly used antileukemic treatments have traditionally included demethylating agents such as azacitidine or decitabine, and low-dose cytarabine [16,17,18,19], or more recently the inhibitor of the B-cell lymphoma-2 (BCL-2) protein, venetoclax [20,21]. 

### 1.2. Leukemic Stem Cells

A blockade of differentiation in AML results in the excess production of undifferentiated and immature leukemic blasts, showing limited proliferative capacity and acquired critical genetic or epigenetic alterations that disrupt important growth-regulatory pathways [22]. AML is expressed as abnormal development in one of the major blood lineages; however, the blasts from patients are heterogeneous with respect to the lineage antigens they express [23,24]. The hierarchical organization within the leukemia clone is also reflected in its cellular heterogeneity, e.g., with respect to morphology, cell surface markers or functional characteristics, and is similar to that of normal hematopoiesis [25].

AML was one of the first diseases for which the existence of a population of cancer stem cells was suggested [26]. Consequently, human AML leukemic stem cells (LSCs) represent one of the most well characterized populations of cancer stem cells [27]. These cells are generally defined by functional assays, i.e., long-term culture or xenograft models, and these functionally defined stem cells can be detected in different cell subsets, i.e., they differ in their expression of cell surface markers [28,29]. 

However, further characterization of the phenotype of LSCs has revealed several differences between LSCs and normal hematopoietic stem cells (HSCs). Most leukemic cells in AML express CD34, which is the normal surface marker for hematopoietic stem/progenitor cells (HSPCs) [30,31]. Within the total AML cell population, LSCs are always a minority, and patients can have AML stem cells with CD34+38-, CD34+CD38+ and/or CD34- phenotypes [32,33]. However, for approximately 25% of AML patients, more than 90% of the total AML cell population is CD34- [32,34,35], and this phenotype is associated with the nucleophosmin 1 (*NPM1*) mutation [33,34]. Finally, the Lin-CD34+ fraction of AML patients can be divided into two subpopulations based on AML stem cells. These two populations are similar to normal lymphoid-primed multipotent progenitors (CD38-CD90-CD45RA+), and granulocyte macrophage progenitors-like cell populations (CD38+CD123+CD45RA+) [35]. In most patients, both populations coexist and possess LSC potential [35]; thus, there is also heterogeneity within the LSC populations.

## 2. Malignant Cell Metabolism and Its Possible Clinical Importance in AML

### 2.1. Metabolism in Malignant Diseases

One of the hallmarks of cancer cells, as proposed by Hanahan and Weinberg in 2000, is their altered metabolic state [36]. The last decades have increased our knowledge of how metabolism is altered in cancer cells, this in concordance with mapping of activation of signal transduction pathways regulating nutrient metabolism. Central for this research is the notion that metabolic pathways are reprogrammed in cancer cells to divert nutrients towards anabolic processes for heightened growth and proliferation. Normally, access to and subsequent utilization of nutrients is highly regulated by physiological cellular signaling mechanisms, providing an important barrier to transformation [37]. Cancer cells can meet their own bioenergetic needs by altering their metabolism to promote growth, survival, proliferation, and long-term maintenance. In normal cells, mitochondria are the main components in the generation and regulation of cellular bioenergetics and are responsible for the main production of ATP by oxidative phosphorylation [37]. Cancer cells do not follow the normal pathway for energy production, and this altered metabolism found in cancer cells includes a switch from mitochondrial oxidative phosphorylation to provide an increased glucose uptake and fermentation of glucose to lactate; this is observed even in the presence of both fully functioning mitochondria and oxygen, and is referred to as the Warburg effect. Metabolism in cancer cells can be reprogrammed from a homeostatic state, with high nutrient catabolism or storage, to an anabolic state where nutrients are transformed into biomass [38].

Switch to aerobic glycolysis has the potential to provide tumor cells with a proliferative advantage, and has thus increased the interest regarding investigation into the mechanisms by which this process is triggered and regulated in order to determine the best means of exploiting this pathway for therapeutic gain [39]. Consequently, given the importance of promoting nutrient uptake and utilization in cell-building processes, it becomes clearer that cancer cells repeatedly select for mutations that increase signal transmission through pathways that unite in a common set of metabolic processes.

### 2.2. Metabolism in AML

As for malignant cells in general, AML cells also have considerable alterations in their metabolic features. Malignant cells often have a reprogrammed and upregulated metabolism with glucose consumption for energy generation, use of glutamine for the refilling of the intermediates of the tricarboxylic acid (TCA) cycle, and fatty acid synthesis for the building of cellular membranes [40], and this has also been shown for AML cells [10,41,42]. AML cells can adapt to meet the increased energy or substrate demand during stressful conditions in the bone marrow microenvironment. However, while normal HSCs apply glycolysis mainly as a source for energy homeostasis, recent studies show that LSCs are mainly dependent on oxidative phosphorylation for survival [43]. In newly diagnosed patients, variable proportions of both high and low oxidative phosphorylation are found in AML cells, whereas in post-chemotherapy patients, there is an excess of high oxidative phosphorylation cells. This indicates that mitochondrial oxidative phosphorylation may be associated with AML chemoresistance, and high oxidative phosphorylation signatures and metabolism are identified as hallmarks of chemoresistance in vivo [43].

Of special interest in the setting of AML and metabolism, is the last decade’s discovery that 10%–15% of AML patients have mutations in the genes isocitrate dehydrogenase (*IDH*)1 and *IDH2* [44]. The IDH proteins are critical for the TCA cycle, catalyzing the oxidative decarboxylation of isocitrate to α-ketoglutarate. Mutations in the *IDH* genes lead to production of the oncometabolite 2-hydroxyglutarate. Hence, a specific metabolic profile associated with *IDH* mutations have been identified, and serum levels of 2-hydroxyglutarate seems to have both potential diagnostic and prognostic impact [45]. Taken together, these findings clearly highlight the importance of metabolic deregulations in supporting leukemia cell survival and growth.

## 3. The Phosphoinositide 3-Kinase (PI3K)-Akt-Mammalian Target of Rapamycin (mTOR) Pathway

### 3.1. Function and Signaling of the PI3K-Akt-mTOR Pathway

The PI3K-Akt-mTOR pathway has been extensively studied in normal and malignant cells [46]. The signaling cascade is activated by a wide variety of extracellular stimuli, including receptor tyrosine kinases, various integrins, B and T cell receptors, and G-protein-coupled receptors (GPCRs). Family members of PI3K are Serine (Ser)/Threonine (Thr) kinase heterodimers, which can be divided into three different classes based on their structural characteristics and substrate specificity [47]. Class I enzymes are separated into class IA and class IB enzymes, both of which are activated by cell surface receptors. Class IA enzymes can be activated by receptor tyrosine kinases (RTKs), GPCRs, and various oncogenes such as the small G protein Ras, whereas class 1B enzymes are activated solely by GPCRs (Figure 1).

Class IA PI3K enzymes include a catalytic (p110) and a regulatory subunit (p85 or p101) [48,49]. In response to extracellular stimuli, recruitment scaffolding proteins, such as the growth factor receptor-bound protein 2 (GRB2)-associated binding protein 2 (GAB2) or insulin receptor substrates (IRS) 1/2, bind to the regulatory p85 subunit of PI3K. Sequentially, the catalytic subunits of PI3K are activated, and phosphorylation of phosphatidylinositol 4,5-bisphosphate (PIP2) generates the second messenger phosphatidylinositol 3,4,5- trisphosphates (PIP3) [50]. This facilitates the recruitment of proteins that contain pleckstrin-homology (PH) domains, including the Ser/Thr kinase Akt (also known as protein kinase B) and its upstream activator 3-phosphoinositide-dependent kinase-1 (PDK1) (Figure 1).

Akt can function as a proto-oncogene, and there are three structurally active forms of Akt in mammalian cells termed Akt1, Akt2, and Akt3 or PKB α, β, γ, respectively [51]. All three isoforms comprise an N-terminal PH domain, a T-loop region of the catalytic domain containing a Thr308 phosphorylation site, and a C-terminal regulatory tail with a Ser473 phosphorylation site [51,52]. Whereas Akt is cytosolic in unstimulated cells, an activation mediated by PI3K requires translocation of Akt to the membrane, where PIP3 serves as an anchor [53]. At the plasma membrane, PDK1 phosphorylates Akt at Thr308, leading to its partial activation. A subsequent phosphorylation at Ser473 is required for full enzymatic activation. This phosphorylation is achieved by the mTOR complex 2 (mTORC2) as well as by members of the PI3K-related kinase (PIKK) family [51,52]. Phosphorylation of homologous residues in Akt2 and Akt3 occurs in the same way. This activation leads to the relocation of Akt to the cytosol or the nucleus, and it has been postulated that Akt then can phosphorylate more than 9000 downstream substrates [54], thereby regulating important cellular processes such as cell metabolism, proliferation, transcription, and survival. The mTOR complex 1 (mTORC1) is an important downstream target of Akt (Figure 1).

The Ser/Thr protein kinase mTOR was first identified in the budding yeast *Saccharomyces cerevisiae* during a trial for resistance to the immunosuppressant drug rapamycin [55]. It belongs to the PIKK family and has a C-terminal catalytic domain with sequence homology to PI3Ks. It is a large, multi-domain protein with protein kinase activity, adding phosphate groups to Ser or Thr residues in a wide range of substrates, many of which are involved in anabolic pathways [56]. Insulin and insulin growth factors (IGFs) [57], nutrients such as amino acids [58], various forms of stress, and the accessibility of ATP are the main regulators of mTOR. 

The activity of mTOR is accomplished by the two separate multi-protein complexes, mTORC1 and mTORC2. The two complexes differ in their protein components, substrate specificity, and regulation, and they have dissimilar responses to rapamycin and its derivatives (rapalogs). The mTORC1 is composed of five units: (i) mTOR, (ii) regulatory-associated protein of mTOR (Raptor), (iii) mammalian lethal with SEC13 protein (mLST), (iv) DEP domain-containing mTOR-interacting protein (Deptor), (V) and the proline-rich Akt substrate of 40 kDa (Pras40) [59] (Figure 1). Both Deptor and Pras40 function as inhibitors. The mTORC2 is composed of mTOR, mLST8, Deptor, the rapamycin insensitive companion of mTOR (Rictor), the mammalian stress-activated protein kinase-interacting protein 1 (mSIN1), and Protor [57,60] (Figure 1).

Raptor is responsible for the phosphorylation of downstream substrates, and this action is blocked by rapamycin. As mTORC2 contains Rictor rather than Raptor, it phosphorylates a different set of substrates, although the regulation of mTORC2 activity remains poorly understood. The mTOR complexes have different upstream mechanisms of activation as well as different downstream substrates. The main downstream targets of mTORC1 are the ribosomal protein S6 kinase (S6K) and eukaryotic initiation factor-4E (eIF4E) -binding protein 1 (4EBP1), while most common substrates of mTORC2 are Akt and related kinases [61]. The S6K protein plays a central yet moderately defined role in cellular and organismal physiology. There are two identified isoforms of S6K, termed p70 and p85, produced by differential splicing from a common gene. Both isoforms are implicated in regulation of cell growth, but the p70S6K isoform has been given the most attention, and the function of the p85S6K remains poorly characterized. Downstream targets of mTORC1 play critical roles in the regulation of translation. The translation initiation complex eIF4F is a heterotrimeric protein complex composed of eIF4E, eIF4A, and eIF4G [61]. The eIF4E binds to the messenger RNA (mRNA) 5’-cap structure to promote the initiation of translation. In the unphosphorylated state, 4EBP1 binds to eIF4E, hindering its association with the complex, and blocking it from binding to mRNA. However, in response to stimuli such as growth factors, mitogens, and amino acids, mTORC1 phosphorylates 4EBP1s, causing it to lose its inhibitory effect. This allows the formation of the eIF4F complex and the subsequent initiation of translation [61]. Furthermore, PDK1 can directly phosphorylate S6K, leading to direct activation of this key regulator of metabolism [62]. 

The FK506-binding protein 38 (FKBP38) is a unique member of the FKBP-family, and acts as an upstream regulator of the PI3K-Akt-mTOR pathway [63]. FKBP38 was identified as an endogenous inhibitor of mTORC1 that binds to the FKBP-C domain of mTOR. This binding interferes with mTOR function in a manner comparable to the FKBP12-rapamycin complex [63]. Under conditions rich in growth factors and nutrients, the FKBP-C domain might interact with Ras homologue enriched in brain (RHEB)-GTP, releasing mTOR from FKBP38 and activating downstream mTOR signaling [63,64]. A negative feedback loop can be formed in the PI3K-Akt-mTOR pathway by activation of mTORC1. Phosphorylated S6K will sequentially phosphorylate IRS proteins, triggering their proteasomal degradation and inhibiting insulin/IGF-1-mediated PI3K activation [65].

The phosphate and homologue protein deleted on chromosome 10 (PTEN) protein and Src Homology 2 (SH2) domain-containing inositol 5-phosphatases (SHIP1 and SHIP2) directly antagonize PI3K by dephosphorylating PIP3 back to PIP2; hence, negatively controlling the PI3K-Akt-mTOR pathway [47] (Figure 1). *PTEN* is the third most commonly mutated gene in human cancers, underlining its functional significance [66,67]. The activity of mTOR is also negatively controlled through the tuberous sclerosis complex (TSC: comprising TSC1 and TSC2). Upon activation, Akt will phosphorylate and inhibit heterodimeric TSC, which then acts as a GTPase-activating protein (GAP) for small GTPase RHEB [68]. This activation leads to hydrolysis of bound GTP with subsequent RHEB inhibition. The inactivation of TSC2 maintains RHEB in its GTP-bound state, thereby supporting augmented activation of mTOR [68]. 

The mTORC1 can also be regulated by cellular stress and energy status through TSC, in addition to growth factors. Moreover, a low energy status can activate AMP-activated protein kinase (AMPK), which in turn phosphorylates Raptor and TSC2, leading to the inhibition of mTORC1 [68,69]. Amino acids regulate mTORC1 in a TSC-independent pathway. Thus, multiple stimuli modulate mTORC1 to control cell growth and autophagy.

### 3.2. The Role of the PI3K-Akt-mTOR Pathway in Modulating Metabolism

Several extracellular stimuli such as insulin and a variety of growth factors are implicated in the PI3K-Akt-mTOR pathway activity [70]. Cell proliferation requires energy, and the PI3K-Akt-mTOR pathway contributes to the regulation of cellular metabolism by several mechanisms [71,72].

Regulation of cellular metabolism by mTOR is complex, as it involves activation of mTORC1 by Akt and upregulation of hypoxia-inducible factor-1 (HIF1), for the promotion of glycolysis by converting pyruvate to ATP molecules and lactate [73]. During physiological oxygen levels, HIF1 is deregulated, but the protein accumulates by increased signaling of mTORC1 and the downstream mediators, 4EBP1 and eIF4E [74]. Furthermore, by promoting translation of nucleus-encoded mitochondria-related mRNAs, mTORC1 can regulate mitochondrial function and oxidative metabolism [38]. mTORC1 also responds to intracellular and environmental stresses, such as low ATP levels, hypoxia, or DNA damage, and participates in regulation of cell growth and metabolism by inducing a shift in glucose metabolism from oxidative phosphorylation to glycolysis [38]. Activation of mTORC1 is reliant on growth factors and amino acids, such as glutamine, leucine, and arginine; this facilitates feedback mechanisms for increased uptake of nutrients to act as fuel for anabolic reactions. However, the activity of mTORC1 is reduced upon starvation [38], and during conditions of high-energy stress, the metabolic regulator AMPK is activated. AMPK represses mTORC1 activity indirectly through phosphorylation and activation of TSC2, and directly through the phosphorylation of Raptor [38].

Interestingly, besides promoting the expression of the enzymes of lipid synthesis and the pentose phosphate pathway, mTORC1 has also been shown to increase expression of the glucose transporter 1 (GLUT1) and other enzymes of glycolysis [74]. Activated mTORC1 also participates in regulation of autophagy and mRNA translation. For example, Poulain and coworkers demonstrated the constitutive activation of mTORC1 in AML cells; hence, indicating the dependence of glucose metabolism in the malignant cells [75]. This implements mTORC1 as one of many contributors to the glycolytic switch found in most cancer cells.

### 3.3. PI3K-Akt-mTOR Signaling in AML

The PI3K-Akt-mTOR pathway is upregulated in AML cells, potentially contributing to metabolic reprogramming, e.g., constitutively active mutant Akt seems to display increased glycolysis [76]. Thus, as a regulator of glucose metabolism, Akt causes upregulation of the glycolysis phenotype. However, no correlation between this increased glycolysis and oxygen consumption rates was observed, suggesting that Akt hyperactivation promotes aerobic glycolysis [76]. In addition, Akt interacts with PDK1, influencing the entrance of pyruvate into the mitochondrial metabolism [77]. Akt may also affect oxidative phosphorylation; it was shown that Akt could promote an indirect oxidative phosphorylation through elevated levels of substrates essential to activity of the TCA cycle and oxidative phosphorylation, such as pyruvate, ADP, and NADH [78].

Dysregulation of the PI3K-Akt-mTOR signaling pathway subsequent to oncogene activating mutations, oncogene amplification, upstream activation of RTKs, or inactivation of tumor suppressor genes has been demonstrated in many human malignancies, including AML [79,80]. The PI3K-Akt-mTOR pathway is central for hematopoietic cells, and regulates crucial functions such as proliferation, differentiation, and survival. This pathway seems to be constitutively activated in 60% of AML patients, and this activation seems to be associated with decreased overall survival [79,80,81,82,83]. Mutations in membrane bound-proteins, such as RTKs or GTPases, are major causes of dysregulated PI3K-Akt-mTOR signaling and are observed in 55% of AML cases [13,66]. However, no available data indicate that PI3K-Akt-mTOR activity is related to specific etiology of the disease, i.e., *de novo*, secondary or refractory/relapsed AML, nor to specific molecular subtypes, i.e., nucleophosmin 1 (*NPM1*) or fms like tyrosine kinase 3 *(FLT3)* mutations or other cytogenetic or molecular genetic alterations [84,85,86].

Signaling initiated through *FLT3* is one of the most important causes of the dysregulation of PI3K-Akt-mTOR signaling in AML, and *FLT3* gene mutations lead to abnormal activation of the pathway [87,88]. Mutations in the *FLT3* gene are among the most frequent mutations seen in AML [89]. These mutations can be classified as internal tandem duplications (*FLT3*-ITD) or mutations in the tyrosine kinase domain (*FLT3*-TKD), with the former being more common in AML [90]. Because *FLT3*-ITDs are located in or near the juxtamembrane domain of the RTK, they affect multiple processes within the activation loop of the TKD, while other mutations are effectively isolated, resulting in the substitution of single amino acids in the loop [90].

The three classes of PI3Ks (Class I-III) have different structure, cellular distribution, mechanism of action, and preference of substrates [91]. Expression of the regulatory p85α subunit of PI3K Class IA was examined in a previous study, which included 40 AML patients, in which 21 patients (53%) demonstrated increased PI3K activation, and PI3K expression correlated with the AML cells proliferation capacity [92]. The catalytic subunits termed p110α, p110β, p110γ, and p110δ are responsible for activation of Akt, and p110δ is the only form that constantly shows elevated expression in human AML [49,93]. 

For most, although not all AML patients, constitutive Akt phosphorylation results from the constitutive activation of the PI3K-Akt-mTOR pathway, which is essential for the survival of AML cells [82,94]. Furthermore, phosphorylation at Thr308 by PDK1 and at Ser473 by mTORC2 constitutively activates Akt, an observation detected among the majority of AML patients [82,95]. Akt phosphorylation at these two sites has been associated with decreased overall survival in several studies [46,82,96], though independently, increased PI3K-Akt activity may constitute a favorable prognostic factor in AML [95]. Finally, aberrant AML cell activation of mTORC1, causing the phosphorylation of downstream targets such as p70S6K, S6RP, and 4EBP1, has also been detected for a large majority of patients [97], but this activation of mTORC1 may not depend on PI3K-Akt activity alone in human AML [98]. The constitutive activation of Akt in AML is supported by autocrine IGF-1/IGF-1R signaling and inhibition of IGF-1R results in decreased activation of Akt for most patients with such autocrine signaling [99]. 

### 3.4. Crosstalk between PI3K-Akt-mTOR Pathway and other Signaling Pathways

Crosstalk between PI3K-Akt-mTOR and other signaling pathways is also important, and among these other pathways, two seem central, namely the Ras-Raf- mitogen-activated protein kinase (MEK) - Ras- extracellular signal-regulated kinase (ERK) and the spleen tyrosine kinase (SYK) pathway. 

The Ras-Raf-MEK-ERK pathway strongly cooperates with the PI3K-Akt-mTOR signaling pathway, exhibit positive and negative regulation of each other, and together regulate several central cellular functions [100,101]. Many mechanisms and methods of crosstalk between the two pathways have been identified, including cross-inhibition, cross-activation, negative feedback loops, and pathway convergence on substrates (Figure 2). 

Binding of an extracellular mitogen to cell surface receptors activates the Ras-Raf-MEK-ERK pathway [101], and activation of the downstream pathway culminates in transcription factors and gene expression stimulation [102]. The Ras-Raf-MEK-ERK- PI3K-Akt-mTOR signaling crosstalk is mediated by three main factors: (i) PI3K, (ii) TSC2, and (iii) mTORC1. Activated, Ras-GTP can directly bind and allosterically activate PI3K [103,104,105]. High activation of the Ras-Raf-MEK-ERK pathway can recruit mTORC1 indirectly through ERK and RSK signaling mediated by the TSC complex [100,106]. The ERK and RSK sites function to promote mTORC1 activity and tumorigenesis, but they are different from those phosphorylated by Akt [100], in that they induce a PI3K-independent mTORC1 phosphorylation of 4EBP1 [100,107,108]. Downstream targets hence involve the Forkhead box O (FOXO) and c-Myc transcription factors, B-cell lymphoma (BCL) 2-associated agonist of cell death (BAD), glycogen synthase kinase 3 (GSK3) and phosphofructokinase-2 (PFK2) (Figure 2). Components of the Ras-Raf-MEK-ERK pathway also exhibit positive regulation of the PI3K-Akt-mTOR pathway. This is mediated by TSC2 and mTORC1 as key integration sites, as they are both affected by PI3K signaling. 

Central intracellular pathways involved in downstream activation of the SYK, also involve the PI3K-Akt-mTOR pathway [109,110]. Chemokine-mediated leukemia support of SYK-dependent modulation of PI3K-Akt-mTOR signaling has been described to be operative in AML [111]. Overexpression of SYK leads to increased activation of mTOR and Ras-Raf-MEK-ERK signaling in AML [109]. This seems to be of special interest in AML patients with mutated *FLT3* and highly activated SYK (112). In an active and phosphorylated state, FLT3 associates with SYK through its C-terminal SH2 domain, and this association increases the activity of FLT3 through phosphorylation. This cooperative activation of FLT3 and SYK results in an increase in the expression of c-Myc and the target genes of c-Myc [112]. The importance of crosstalk between SYK and PI3K-Akt-mTOR signaling is further supported by the observation that inhibition of PI3K-Akt-mTOR pathway activity enhances the effects of SYK inhibition on AML cell differentiation and viability [112,113]. Subsequent to an activation of the receptor complex Src, an active Src phosphorylates SYK at tyrosine residues, leading to its activation [114]. While active, SYK phosphorylates p110/p85 subunits of PI3K, and hence increasing its catalytic activity, which converts PIP2 to PIP3 [114]. Chemical inhibition of SYK has been tested in multiple cells lines and results in a dose-dependent inhibition of mTOR [109]. Furthermore, in some AML cell lines and primary cells, both p70S6K and 4EBP1/eIF4E signaling pathways are influenced by manipulations in SYK activity, and combined chemical inhibition of SYK and eIF4E has shown a synergistic effect on AML viability in multiple AML cell lines [109]. This is supported by inhibition of eIF4E through SYK knockdown, indicating that there are enhanced inhibitory effects of SYK on cell viability. Inhibition of eIF4E enhanced AML differentiation when combined with SYK inhibition using either chemical or SYK-directed short hairpin RNA [109].

## 4. Inhibition of the PI3K-Akt-mTOR Pathway in AML

It is commonly accepted that a disparity in the regulation of protein kinases may be one of the primary causes of genetic-based human diseases, and a major part of new drug development over the last decades has been dedicated to the development of protein kinase inhibitors [115]. The PI3K-Akt-mTOR pathway has emerged as a possible therapeutic target in human malignancies, and several pharmacological inhibitors have been developed, including isoform-selective or pan-class I PI3K inhibitors, Akt inhibitors, rapamycin, and rapalogs as well as dual PI3K-mTOR inhibitors [116]. The rapalogs everolimus and temsirolimus represent allosteric inhibitors directed towards mTORC1 and are now in clinical use. Studies have shown that PI3K positively controls the phosphorylation of Bad and FOXO3A (Figure 2), two targets downstream of Akt that control cell survival, although, specific inhibition of PI3K does not induce significant apoptosis in primary AML cells [117].

The efficiency of several selective inhibitors towards the PI3K-Akt-mTOR pathway has been studied in AML cell lines and primary AML cells, and results have shown that Akt can be reactivated through negative feedback mechanisms subsequent to rapamycin-mediated mTOR inhibition. Inhibitors targeting the catalytic domain of mTOR have also been developed [118], and pre-clinical studies of rapamycin combined with rapalogs targeted at mTOR showed inhibition of clonogenic AML cell proliferation without inhibition of normal CD34^+^ cells [97]. The antileukemic effects were further enhanced by combination with conventional cytotoxic drugs [119]. However, the relevant antileukemic activity of rapamycin and rapalogs has not been demonstrated in clinical trials [120,121,122,123,124,125]. 

Both PI3K and mTOR are members of the PIKK superfamily and share structural domains, and some inhibitors therefore act on both kinases. An important consequence of inhibiting mTORC1/S6K by rapalogs has been the increased phosphorylation of Akt [118,126]; this has been observed in both experimental and clinical studies. Dual inhibition of PI3K and mTOR blocks both the upstream and downstream pathways of Akt, consequently circumventing activation of Akt subsequent to the reduced mTORC1-S6K-IRS1 mediated negative feedback loop [127]. Possible reasons for the failure to demonstrate clinically relevant antileukemic effects by pathway inhibitors include limited activity in human patients [128], intrinsic molecular defects [129], and dose-limiting toxic effects [130]. Akt is important for a wide range of cellular functions and interacts with an immense number of substrates [52]. Only a few Akt inhibitors have been developed, and they have not produced convincing effects in AML [131,132]. Results from selected representative and relatively large clinical studies of PI3K-Akt-mTOR inhibitors were recently reviewed by Herscbein and Liesveld [133], and Table 1 represents the most relevant studies. These clinical studies justify the conclusions that (i) pathway inhibitors used alone have only modest antileukemic effects; (ii) pathway inhibition in combination with intensive chemotherapy have acceptable toxicity; (iii) the doses used in clinical studies can alter the activation of pathway mediators; and (iv) the most important toxicities revealed by clinical studies are hematological and gastrointestinal toxicity. However, while there is some evidence it may be more effective in certain subsets of patients, clinical experience with mTOR inhibition is extremely limited and has not produced convincing results.

## 5. Conclusions and Further Perspective

AML patients harbor leukemia cells that display heterogeneous constitutive PI3K-Akt-mTOR activation profiles [79,95]. These dissimilarities are part of more complex phenotypic variations, including transcriptional regulation, cross signaling, and cell communication and may be influenced by growth factor stimulation or various inhibitors of pathway activation [70,79,80]. This is probably again reflected in the heterogeneity between leukemic cells with regard to energy metabolism, amino acid metabolism and arachidonic acid metabolism, supported by the finding that modulation of arachidonic acid metabolism alters the activation of mTOR and its downstream mediators [80]. The activation status of the PI3K-Akt-mTOR pathway is probably also reflected in clonal heterogeneity and different gene expression profiles [79]. The PI3K-Akt-mTOR pathway is important for disease development and chemosensitivity, and activation of this pathway has prognostic impact in AML [79,95]. However, the results from the initial clinical studies using PI3K/mTOR inhibitors are regarded as disappointing by many clinicians [100], one should however keep in mind that results in relapsed/refractory patients may not be representative of effects in patients treated at the time of first diagnosis. Biological identification of patient subsets should be evaluated in future clinical studies using this therapeutic strategy. Given the limited effects of these inhibitors so far in clinical trials, it appears to be more promising to combine PI3K-Akt-mTOR inhibitors with other pharmacological agents. Several new approaches have emerged the last years in the treatment of AML [134,135]; many of these have already been established in treatment or have entered clinical trials. Of special interest in the setting of PI3K-Akt-mTOR signaling and inhibition, is the entrance of inhibitors of the SYK- and Ras-Raf-MEK-ERK pathways. In Table 2, we summarize both MEK and SYK inhibitors, as well as the major new drugs that have emerged as potential agents for AML treatment, and shortly address the potential benefits in combining them with PI3K-Akt-mTOR inhibitors. Given the critical roles of SYK in multiple cellular functions, the interest in the identification of SYK inhibitors has seen a sharp increase in patent filings claiming such compounds [136]. Some of these SYK inhibitors have been investigated in clinical trials including patients with autoimmune diseases and hematological malignancies [111,117], although several of the new and more potent inhibitors have so far not been evaluated in clinical trials. An interest in inhibition of the Ras-Raf-MEK-ERK pathway has also increased the last decade [137], and several inhibitors entering preclinical and clinical trials have emerged. MEK is probably the target where development has reached the farthest, and recently a clinical trial with the MEK inhibitor binimetinib was published [138]. Although the antileukemic effects of this inhibitor in monotherapy seem limited, this should further encourage investigating Ras-Raf-MEK-ERK in combination therapy, including inhibitors of the PI3K-Akt-mTOR pathway in AML [137,139]. 

The numbers of mutant signaling proteins identified in AML are high [9], although are more likely to reflect activation of a limited number of downstream effector pathways [115]. Targeting of these unifying pathways hence may represent a more broadly applicable therapeutic strategy, emphasizing the potential rational for combination therapy in AML [139]. Combined treatment, e.g., the combination of different pathway inhibitors (i.e., dual pathway inhibition) and the combination of pathway inhibitors with other targeted therapies or conventional chemotherapy should therefore also be considered in clinical AML trials. Although the contributions of nearly all components of the PI3K-Akt-mTOR pathway is currently described in mechanistic detail, there are issues that need to be addressed further, such as cooperation of this pathway with other pathways. Thus, further analyses of the interrelations of this pathway with other signaling mediators, pathways and/or genetic factors are hence needed.

Furthermore, the AML heterogeneity is also reflected by the different effects of PI3K-Akt-mTOR inhibition, and as also demonstrated by our research group, the effects of these inhibitors seem to vary considerably among AML patients [70,79,80,140]. No clear mutational profile or other pathological process related to the disease, has so far been detected to predict response to treatment, although these features must be evaluated in larger in vitro and in vivo studies. 

## Figures and Tables

**Figure 1 ijms-21-02907-f001:**
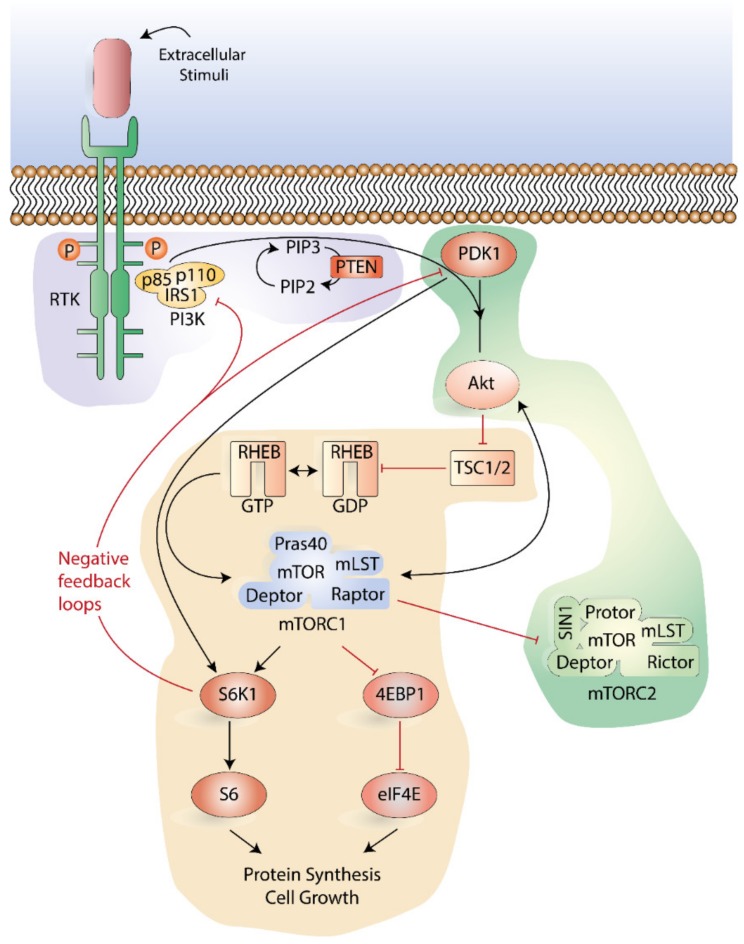
Overview of the phosphoinositide 3-kinase-Akt-mammalian target of rapamycin (PI3K-Akt-mTOR) signaling pathway. Following ligation of cell surface receptors (e.g., growth factor receptors) phosphorylated receptor tyrosine kinases (RTK) recruits scaffolding proteins, which bind to the regulatory p85 subunit of PI3K. A subsequent activation of the catalytic subunits of PI3K generates phosphatidylinositol 3,4,5- trisphosphates (PIP3). Phosphoinositide-dependent kinase-1 (PDK1) and Akt proteins are then recruited to the plasma membrane, inducing the phosphorylation of Akt on Thr308 by PDK1. This is followed by activation of Akt on Ser473 by the mTOR complex 2 (mTORC2); this second phosphorylation is necessary for complete activation. Akt controls the activation of mTOR complex 1 (mTORC1) by constraining the GTPase activity of the TSC1/TSC2 complex towards the Ras-related GTP-binding protein ras homologue enriched in brain (RHEB) that associates to mTORC1 and phosphorylates mTOR. The mTORC1 induces cap-dependent messenger RNA (mRNA) translation by phosphorylating 4EBP1, leading to the formation of eIF4F and the inhibition of autophagy. Both mTORC1 and PDK1 can directly activate S6K1, which in turn activates S6, and hence facilitates protein synthesis and cell growth. Positive regulation (activation/stimulation) of the pathways is presented as black arrows, and negative regulation (inhibition) of the pathways is presented as red blunt-ended lines. The abbreviations shown in the figure can be found in the list of abbreviations.

**Figure 2 ijms-21-02907-f002:**
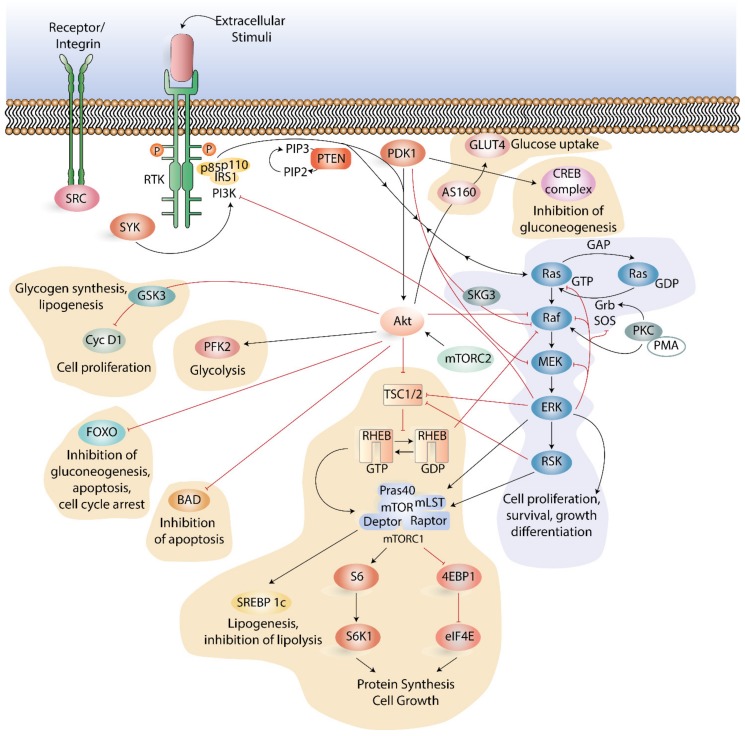
Main components and crosstalk between the Ras-Raf-mitogen-activated protein kinase-Ras- extracellular signal-regulated kinase (Ras-Raf-MEK-ERK) pathway, the spleen tyrosine kinase (SYK) pathway and the PI3K-Akt-mTOR pathway. All pathways respond to extracellular and intracellular signals to control cell survival, proliferation, motility, and metabolism, and are activated by the binding of a growth factor to a receptor tyrosine kinase (RTK) or integrins to receptors. In the Ras-MAPK pathway, this binding produces binding sites for the SHC and GRB2 adaptor molecules that recruit SOS to the membrane. SOS catalyzes the exchange of Ras GDP to Ras GTP, followed by the recruitment and activation of Raf. Protein kinase C (PKC) is directly bound and activated by phorbol 12-myristate 13-acetate (PMA) through its mimicking of the natural ligand of PKC. The mechanism by which PKC activates ERK remains unclear. Following a Ras activation of MEK, ERK is activated through an activation loop phosphorylation. The pathway is further regulated by a negative feedback loop from ERK. Pathway crosstalk is regulated through cross-inhibition and cross-activation between the components of the Ras-Raf-MEK-ERK and PI3K-Akt-mTOR pathways. Pathway crosstalk can also be regulated by binding of the p85 subunit of PI3K by phosphorylated SYK. When active, SYK increases the catalytic activity of PI3K, resulting in conversion of phosphatidylinositol 4,5-bisphosphate (PIP2) to PIP3. PI3K-Akt and Ras-Raf-MEK-ERK signaling networks exhibit examples of crosstalk between intracellular signaling proteins. Positive regulation (activation/stimulation) of the pathways is presented as black arrows, and negative regulation (inhibition) of the pathways is presented as red blunt-ended lines. PI3K and Ras-Raf-MEK-ERK signaling meet on mTORC1, which is the master controller of protein translation. PTEN acts as a strong negative regulator of both pathways. Akt can cross-react with among others the Forkhead box O (FOXO), B-cell lymphoma (BCL) 2-associated agonist of cell death (BAD), glycogen synthase kinase 3 (GSK3) and phosphofructokinase-2 (PFK2). The abbreviations shown in the figure can be found in the list of abbreviations.

**Table 1 ijms-21-02907-t001:** Important clinical studies with mTOR inhibitors. The table summarizes the most important findings in larger studies (>20 patients included) using mTOR inhibitors either alone or in combination with other chemotherapy regimens for AML patients. The abbreviations shown in the table can be found in the list of abbreviations.

Study	mTOR Small-Molecule Inhibitor	Patients	Treatment	Summary of Results	Toxicity/ Major Side Affects
Rizzieri et al [122]	Ridaforolimus (also known as AP23573, MK-8669, or Deforolimus)	55 patients, 23 patients with AML and three with other myeloid malignancies	Ridaforolimus 12.5 mg intravenous infusion for 5 days every 2 weeks	No complete remissions (CR) or partial remissions ( PR) Stable disease for a minority of patients	Mouth sores Fatigue Nausea Thrombocytopenia
Perl et al [120]	Sirolimus (also known as Rapamycin)	29 patients with refractory or relapsed AML	Sirolimus in a 12 mg loading dose on day 1 followed by 4 mg/d on days 2 to 7, in parallel with chemotherapy.	CR or PR in 6 (22%) of the 27 patients who completed chemotherapy	Marrow aplasia Multi organ failure
Park et al. [124]	Everolimus (also known as RAD001)	28 AML patients below 65 years of age in first relapse.	Everolimus in increasing doses from 10 to 70 mg, administrated orally on days 1 and 7 in combination with conventional 3 + 7 daunorubicin + cytarabine induction therapy.	CR in 68% of patients. Subsequent intensification with allogeneic stem cell transplantation in 29% of patients	Gastrointestinal Respiratory
Amadori et al. [125]	Temsirolimus (also known as CCI-779)	53 patients with primary refractory or first relapse AML	Clofarabine 20 mg/m^2^ on days 1–5 and temsirolimus 25 mg on days 1, 8, and 15 If CR or CRi- monthly temsirolimus maintenance therapy	CR in 8% of patients CRi in 13% of patients Median DFS 3.5 months. Median OS 4 months (9.1 months for responders)	Infectious complications Febrile neutropenia Transaminitis

**Table 2 ijms-21-02907-t002:** Emerging therapeutic interventions in AML and their potential benefits in combination with PI3K-Akt-mTOR inhibitors. The table summarizes new drugs in AML therapy, and shortly describes the potential benefits of combining them with PI3K-Akt-mTOR inhibitors. Key references for the new agents and their use in AML are given in the table.

Targets	Potential Agents	Potential Advantages in Combination with PI3K-Akt-mTOR Inhibitors	Key References
DNA methylation	Azacitidine, decitabine	Potential synergism through the increase of Akt suppression and the promotion of mTOR inhibitor expression such as PTEN	[141,142]
BCL-2	Venetoclax	Potential to inhibit AML cell growth	[143]
SYK	Fostamatinib	As SYK cross-reacts with the PI3K-Akt-mTOR, it may be a more broadly applicable therapeutic strategy	[136]
MEK	Binimetinib	Inhibition of both Ras-Raf-MEK-ERK and PI3K-Akt-mTOR pathways and their crosstalk can decrease signaling activity in both pathways, especially in RAS mutated cases	[138]
CXCR4/CXC12	Plerixafor	CXCR4 antagonist can lead to sensitization for both conventional chemotherapy and signaling cascade inhibitors	[144]
FLT3	Midostaurin, gilteritinib	Dual inhibition of FLT3 activation and downstream intracellular targets may potentially have synergistic effects, especially in *FLT3* mutated cases	[145,146]
CD33	Gemtuzumab ozogamicin	Inhibition of extracellular binding and signaling can potentiate the effect of PI3K-Akt-mTOR inhibition	[147]
IDH1	Ivosidenib	Potentiates the alterations in metabolism associated with PI3K-Akt-mTOR, especially in *IDH*1 mutated cases	[148]
IDH2	Enasidenib	Potentiates the alterations in metabolism associated with PI3K-Akt-mTOR, especially in *IDH2* mutated cases	[149]

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
