# Peer review of "The PI3K-Akt-mTOR Signaling Pathway in Human Acute Myeloid Leukemia (AML) Cells"

_ijms, 2020, doi:10.3390/ijms21082907_

Round 1

Reviewer 1 Report

The manuscript is a review of PI3K-Akt-mTOR pathway in the context of acute myeloid leukemia (AML). It reports the importance of cellular metabolism in malignant cells, the role of oxidative phosphorylation in AML chemoresistance and metabolic deregulation caused by IDH mutations in this pathology. Actors of the PI3K-Akt-mTOR signaling pathway are described and the dysregulation of this pathway in AML is documented. The crosstalk of this pathway with the Ras-Raf-MEK-ERK pathway is illustrated to explain cross-inhibitions and cross-activations described in the literature. Finally, the PI3K-Akt-mTOR pathway inhibitors used in clinical studies are discussed.

The manuscript structure is coherent but there is a big problem of clarity and synthesis in many parts of the text. Moreover, figures are not clear and their legends are not always in accordance with the illustrations. I recommend extensive rewriting of the manuscript.

Major points:

The lack of synthesis and precision of many parts of the manuscript leaves a confusing message in the text, the figures and the table.

1/ in the text, there are numerous repetitions inside paragraphs or sections highlighting a lack of synthesis, for example:

page 7 (part 3.2) : At beginning of this part, “PI3K-Akt-mTOR pathway contributes to the regulation of cellular metabolism by upregulation of glucose transporters” (lanes 13-14) is mentioned and later in this part, “mTORC1 has also been shown to increase expression of GLUT1” (lane 31).

page 7 (lanes 30-31/33):promoting the expression of the enzymes of lipid synthesis and the pentose phosphate pathway” and “Activated mTORC1… lipid synthesis, glycolysis and the pentose phosphate pathway”.

lanes 46-47In addition to its effect on glycolysis, Akt may also affect oxidative phosphorylation; it was shown that Akt could promote an indirect oxidative phosphorylation”.

There are also inaccuracies creating confusion and incomplete messages, for example:

page 6 (lanes 32-34):In addition, PDK1 and mTORC1 phosphorylates the S6 kinase, which in turn phosphorylates numerous substrates that are involved in translation”. S6K activation by mTORC1 is already described upstream. In contrast, for PDK1, there are neither explanations nor references.

page 9 (lanes 20-24): ERK and AGC kinases are mentioned to act on FOXO, BAD, GSK3 and PFK2 while in figure 2 legend, it is noted that “Akt can cross react with among others the FOXO, BAD, GSK3 and PFK2”. This part has to be corrected.

page 6 (lanes 35-43): this part refers to figure 1 but nothing is shown in the figure. Authors have to be in accordance with the illustration.

2/ in the figures, there are errors in sequence of pathway activation, for example:

page 5 (figure 1): mTORC1 activates S6K1 and the main target of S6Ks is ribosomal protein S6, a component of the 40S ribosome subunit. The figure shows activation of S6K1 by S6!

Legends are either incomplete or overdeveloped, for example:

page 5 (figure 1): there is no explanation either on S6K activation or on negative feedback loops. Authors have to comment as they did for the 4EBP1 pathway.

page 10 (figure 2): lanes 13-14: “Each pathway has a mechanism to negatively feed onto the other: ERK phosphorylation of GAB connected to PI3K and Akt phosphorylation of Raf”. GAB should be in the figure or authors have to remove the sentence.

lanes 14-16: this explanation has to be in the text and not in the legend.

3/ in table 1 (page 12-13), the four clinical studies described have some disparate information. Globally, authors have to harmonize studies with each other, i.e. give the same information for all studies.

The content of the columns have to be completely redesigned, for example:

  • Column “Patients and treatment”: it’s better for the clarity of information to share in two columns: 1/ patients and 2/ treatments.
  • Column “Summary of results” does not contain a summary at all. For example, in Park et al study, “Complete remission in 68% percent of patients” is specified at the beginning of the results and at the end a variant of this information is given: “Complete remission rates in patients with everolimus areas under or above the curve median were 53% versus 85%”. Authors have to synthetize results by making a list of main results but not sentences.
  • Column “Toxicity”: this part is only to describe secondary effects of treatment. It’s not necessary to indicate again the treatment dose like in Perl et al study: “The maximum tolerated dose of sirolimus; 12 mg loading dose on day 1 followed by 4 mg/d on days 2 to 7, concurrent with MEC chemotherapy” which is already stated in treatment column.

Minor points:

1- page 8 (lanes 32-33):though independently, phosphorylation at Ser473 may constitute a favorable prognostic factor (85)”. In the reference, authors didn’t demonstrate a correlation between Ser473 phosphorylation and prognosis, but constitutive PI3K/AKT activity is a favorable prognosis factor in de novo AML patients without distinguishing which phosphorylation site on AKT is implicated. So, authors have to substitute Ser473 by PI3K/AKT activity.

2- page 5 (figure 1): there is an error in mTORC2 complex. Pras40 is specific of mTORC1. So, authors have to change Pras40 by Protor in mTORC2 complex.

3- There are a lot of typing errors and some grammatical mistakes, for example:

  • page 4 (lane 28):Phosphorylation of homologous residues in Akt2 and Akt3 occur in the same way”.
  • page 6 (lanes 20-21) : used TORC1 or mTORC1 and TORC2 or mTORC2
  • page 8 (lane 47):While some of the kinases, such as. Raf, MAPK ERK kinase…”
  • page 9 (Lane 6): “Cell surface receptors activates…”
  • page 9 (lane 18): “The ERK and RSK sites functions…”
  • page 11 (lane 12): “(Error! Reference source not found.)”
  • pages 8-10: be consistent in the name of pathway Ras MAPK or Ras ERK or Ras-Raf-MEK-ERK
  • Page 12 (Table 1): “Summary ff results”

4- Some abbreviations have to be added: mSIN1, SREBP12, CREB, AGC.

Author Response

Reviewer 1

The manuscript is a review of PI3K-Akt-mTOR pathway in the context of acute myeloid leukemia (AML). It reports the importance of cellular metabolism in malignant cells, the role of oxidative phosphorylation in AML chemoresistance and metabolic deregulation caused by IDH mutations in this pathology. Actors of the PI3K-Akt-mTOR signaling pathway are described and the dysregulation of this pathway in AML is documented. The crosstalk of this pathway with the Ras-Raf-MEK-ERK pathway is illustrated to explain cross-inhibitions and cross-activations described in the literature. Finally, the PI3K-Akt-mTOR pathway inhibitors used in clinical studies are discussed.

The manuscript structure is coherent but there is a big problem of clarity and synthesis in many parts of the text. Moreover, figures are not clear and their legends are not always in accordance with the illustrations. I recommend extensive rewriting of the manuscript.

Major points:

The lack of synthesis and precision of many parts of the manuscript leaves a confusing message in the text, the figures and the table.

1/ in the text, there are numerous repetitions inside paragraphs or sections highlighting a lack of synthesis, for example:

page 7 (part 3.2) : At beginning of this part, “PI3K-Akt-mTOR pathway contributes to the regulation of cellular metabolism by upregulation of glucose transporters” (lanes 13-14) is mentioned and later in this part, “mTORC1 has also been shown to increase expression of GLUT1” (lane 31).

page 7 (lanes 30-31/33):promoting the expression of the enzymes of lipid synthesis and the pentose phosphate pathway” and “Activated mTORC1… lipid synthesis, glycolysis and the pentose phosphate pathway”.

lanes 46-47In addition to its effect on glycolysis, Akt may also affect oxidative phosphorylation; it was shown that Akt could promote an indirect oxidative phosphorylation”.

There are also inaccuracies creating confusion and incomplete messages, for example:

page 6 (lanes 32-34):In addition, PDK1 and mTORC1 phosphorylates the S6 kinase, which in turn phosphorylates numerous substrates that are involved in translation”. S6K activation by mTORC1 is already described upstream. In contrast, for PDK1, there are neither explanations nor references.

page 9 (lanes 20-24): ERK and AGC kinases are mentioned to act on FOXO, BAD, GSK3 and PFK2 while in figure 2 legend, it is noted that “Akt can cross react with among others the FOXO, BAD, GSK3 and PFK2”. This part has to be corrected.

page 6 (lanes 35-43): this part refers to figure 1 but nothing is shown in the figure. Authors have to be in accordance with the illustration.

We agree that the description of PI3K-Akt-mTOR pathway and its association to metabolism was unclear and somehow confusing described in the text. This was also the fact for PDK1 and S6 Kinase activation, and the association to ERK. Accordingly, we have restructured this in our manuscript, special avoiding unnecessary repetitions. We have also improved Figure 1 to better illustrate these facts. Conflicting presentations are clarified and currently more structured and in accordance with figures/figures legends in our revised manuscript. 

2/ in the figures, there are errors in sequence of pathway activation, for example:

page 5 (figure 1): mTORC1 activates S6K1 and the main target of S6Ks is ribosomal protein S6, a component of the 40S ribosome subunit. The figure shows activation of S6K1 by S6!

Legends are either incomplete or overdeveloped, for example:

page 5 (figure 1): there is no explanation either on S6K activation or on negative feedback loops. Authors have to comment as they did for the 4EBP1 pathway.

page 10 (figure 2): lanes 13-14: “Each pathway has a mechanism to negatively feed onto the other: ERK phosphorylation of GAB connected to PI3K and Akt phosphorylation of Raf”. GAB should be in the figure or authors have to remove the sentence.

lanes 14-16: this explanation has to be in the text and not in the legend.

 We appreciate the comments regarding the imprecisely annotations in the figures and the text. Accordingly, we have revised Figure 1 to more clearly demonstrated the activation of S6, and we have restructured the text to more clearly present this part of the manuscript.

3/ in table 1 (page 12-13), the four clinical studies described have some disparate information. Globally, authors have to harmonize studies with each other, i.e. give the same information for all studies.

The content of the columns have to be completely redesigned, for example:

  • Column “Patients and treatment”: it’s better for the clarity of information to share in two columns: 1/ patients and 2/ treatments.
  • Column “Summary of results” does not contain a summary at all. For example, in Park et al study, “Complete remission in 68% percent of patients” is specified at the beginning of the results and at the end a variant of this information is given: “Complete remission rates in patients with everolimus areas under or above the curve median were 53% versus 85%”. Authors have to synthetize results by making a list of main results but not sentences.
  • Column “Toxicity”: this part is only to describe secondary effects of treatment. It’s not necessary to indicate again the treatment dose like in Perl et al study: “The maximum tolerated dose of sirolimus; 12 mg loading dose on day 1 followed by 4 mg/d on days 2 to 7, concurrent with MEC chemotherapy” which is already stated in treatment column.

We highly appreciate these comments, and accordingly we have restructured Table 1, making it easier for readers to follow and interpretant.

Minor points:

1- page 8 (lanes 32-33):though independently, phosphorylation at Ser473 may constitute a favorable prognostic factor (85)”. In the reference, authors didn’t demonstrate a correlation between Ser473 phosphorylation and prognosis, but constitutive PI3K/AKT activity is a favorable prognosis factor in de novo AML patients without distinguishing which phosphorylation site on AKT is implicated. So, authors have to substitute Ser473 by PI3K/AKT activity.

2- page 5 (figure 1): there is an error in mTORC2 complex. Pras40 is specific of mTORC1. So, authors have to change Pras40 by Protor in mTORC2 complex.

3- There are a lot of typing errors and some grammatical mistakes, for example:

  • page 4 (lane 28):Phosphorylation of homologous residues in Akt2 and Akt3 occur in the same way”.
  • page 6 (lanes 20-21) : used TORC1 or mTORC1 and TORC2 or mTORC2
  • page 8 (lane 47):While some of the kinases, such as. Raf, MAPK ERK kinase…”
  • page 9 (Lane 6): “Cell surface receptors activates…”
  • page 9 (lane 18): “The ERK and RSK sites functions…”
  • page 11 (lane 12): “(Error! Reference source not found.)”
  • pages 8-10: be consistent in the name of pathway Ras MAPK or Ras ERK or Ras-Raf-MEK-ERK
  • Page 12 (Table 1): “Summary ff results”

4- Some abbreviations have to be added: mSIN1, SREBP12, CREB, AGC.

The minor points as stated by the reviewer are of significant important, and accordingly we have rewrite/rephrase all this paragraphs in our revised manuscript. The abbreviation list is extended.

Reviewer 2 Report

In their paper Nepstad and others are reviewing the importance of The PI3K-Akt-mTOR signaling pathway in human AML in particular with an eye on the use of inhibitors in the patients and suggesting the use of this pathway as a possible biomarker in preclinical and clinical studies.

The paper is well written and provides interesting information with regards to the importance of the PI3K-Akt-mTOR signalling pathway in AML referring to the studies performed so far.

The authors, in paragraph 3 (3.2/3.2/3.4), provide very fine details about the PI3K-Akt-mTOR signalling pathway and crosstalk with other pathways (eg RAS). This is very interesting, but it fall outside from the scope of the review. I would recommend the authors to highlight the most important points in these sections with relation to the AML pathway and provide a more spot on paragraph with regards of the topic of their review and the importance of these pathways and crosstalk in AML.

Minor: In their title they use the term “signaling” instead of signalling. This is a common American use of the term in contrast to the British English. It is not a mistake but “signalling” would be more appropriate.

Author Response

In their paper Nepstad and others are reviewing the importance of The PI3K-Akt-mTOR signaling pathway in human AML in particular with an eye on the use of inhibitors in the patients and suggesting the use of this pathway as a possible biomarker in preclinical and clinical studies.

The paper is well written and provides interesting information with regards to the importance of the PI3K-Akt-mTOR signaling pathway in AML referring to the studies performed so far.

The authors, provide very fine details about the PI3K-Akt-mTOR signalling pathway and crosstalk with other pathways (eg RAS). This is very interesting, but it fall outside from the scope of the review. I would recommend the authors to highlight the most important points in these sections with relation to the AML pathway and provide a more spot on paragraph with regards of the topic of their review and the importance of these pathways and crosstalk in AML.

In the revised manuscript we have reduced the size of the sections describing crosstalk with other pathways, however we believe some part of this description is also important, and therefore we have restructured this section and made it more easy for the readers to follow.

 Minor: In their title they use the term “signaling” instead of signalling. This is a common American use of the term in contrast to the British English. It is not a mistake but “signalling” would be more appropriate.

We apricate this comment and has consequently used “siganlling” and not “signaling” in our revised manuscript.

Reviewer 3 Report

The review is easy to read and demonstrates clarity. The pathways are well described and illustrated.

However, it does not address the molecular heterogeneity with respect to the PI3K-Akt-Mtor outside of FLT ITD. Description from their paper in Cancers could be added and described Cancers (Basel). 2018 Sep 14;10(9). pii: E332. doi: 10.3390/cancers10090332.  The interesting part is that paper describes secondary Aml. Is there a difference on PI3k dependency between denovo and relapsed AML versus secondary AML? Are their particular molecular subtypes that have PI3K/akt Mtor dependency?

Many of the references are also reviews and could be updated based on a new papers. Descriptions of the recent papers by this group in detail regarding the experiments with insulin and PI3K inhibitors differences and how this information may contribute to the combination therapy  design should be discussed.

Line 24 discussion section of other Signalling pathways and Mtor  signalling should include SYK. See references below.

Leukemia. 2013 Nov;27(11):2118-28. doi: 10.1038/leu.2013.89. Epub 2013 Mar 28.

Cancer Cell. 2014 Feb 10;25(2):226-42. doi: 10.1016/j.ccr.2014.01.022.   These papers speak to the role of SYK in MTorr regulation which may be critical when discussing combination therapy in the last paragraph.   Given that there are many small molecules targeting all these pathways: SYK, MTOR, PI3K and MEK/ERK a discussion on their success and if they have made it to clinical trial should be added. If they have failed this should be discussed. it mentions a separate review for the clinical side and reports in table 1 on MTOR inhibitors many in clinical trials in the 2009-2013 era.    A section on how the groups discoveries could improve upon the current knowledge would be helpful as they have a lot important work that could be important for the design and development of combination treatments to treat this deadly disease.   In patients ineligible for induction therapy novel agents such as venetoclax or hypomethylating agents . How do these drugs affect the PI3K axis and your current work.   This review should bring the reader some context on how your work integrates with modern therapy for AML and how rapamacyin or other like inhibitors could be repurposed in the future.    

Author Response

The review is easy to read and demonstrates clarity. The pathways are well described and illustrated.

However, it does not address the molecular heterogeneity with respect to the PI3K-Akt-Mtor outside of FLT ITD. Description from their paper in Cancers could be added and described Cancers (Basel). 2018 Sep 14;10(9). pii: E332. doi: 10.3390/cancers10090332.  The interesting part is that paper describes secondary Aml. Is there a difference on PI3k dependency between denovo and relapsed AML versus secondary AML? Are their particular molecular subtypes that have PI3K/akt Mtor dependency?

We are glad for these comments related to our previously work. As far as our knowledge, data from our groups and other available data have not proven any association between PI3K-Akt-mTOR activity and specific disease etiology, i.e. de novo, secondary or refractory/relapsed AML, nor to specific molecular subtypes, i.e. NPM1 or FLT3 mutations or other cytogenetic or molecular genetic alterations. However, given the heterogeneity within the AML filed, and the difficulties to study many cases, it is not impossible that such a correlation exist. We have discussed these features in our revised version, together with references to recently publications.

Many of the references are also reviews and could be updated based on a new papers. Descriptions of the recent papers by this group in detail regarding the experiments with insulin and PI3K inhibitors differences and how this information may contribute to the combination therapy design should be discussed.

Accordingly, we have in our revised manuscript included several new and relevant references in the manuscript. 

Line 24 discussion section of other Signalling pathways and Mtor  signalling should include SYK. See references below.

Leukemia. 2013 Nov;27(11):2118-28. doi: 10.1038/leu.2013.89. Epub 2013 Mar 28.

Cancer Cell. 2014 Feb 10;25(2):226-42. doi: 10.1016/j.ccr.2014.01.022.   These papers speak to the role of SYK in MTor regulation which may be critical when discussing combination therapy in the last paragraph.   Given that there are many small molecules targeting all these pathways: SYK, MTOR, PI3K and MEK/ERK a discussion on their success and if they have made it to clinical trial should be added. If they have failed this should be discussed. it mentions a separate review for the clinical side and reports in table 1 on MTOR inhibitors many in clinical trials in the 2009-2013 era.    A section on how the groups discoveries could improve upon the current knowledge would be helpful as they have a lot important work that could be important for the design and development of combination treatments to treat this deadly disease.   In patients ineligible for induction therapy novel agents such as venetoclax or hypomethylating agents . How do these drugs affect the PI3K axis and your current work.   This review should bring the reader some context on how your work integrates with modern therapy for AML and how rapamacyin or other like inhibitors could be repurposed in the future.    

We are grateful for these comments from the reviewer. Given the limited effects of these inhibitors so far in clinical trials, it seems reasonable to combine PI3K-Akt—mTOR inhibition with other pharmacological agents. Therefore, we have included a new Table 2 there we discuss these potential benefits of combining PI3K-Akt—mTOR inhibition with more recently developed pharmacological agents, special emphasizing their potential role in combination with agents targeting PI3k-AKT-mTOR. We have also discussed this features in light of our recent studies.

Round 2

Reviewer 1 Report

Major corrections have been made in the manuscript and recommendations have been taken into consideration by the authors.

However, a problem of clarity and synthesis persists in the parts of the text that have been rewritten. Despite adding a new table (table 2) listing new drugs used in AML treatment, the potential benefits of combining them with PI3K-Akt-mTOR inhibitors need to be completely re-discussed.

I recommend rewriting some parts of the manuscript to clarify the message.

Major points:

page 9: In paragraph 3.4, the authors described a crosstalk between PI3K-Akt-mTOR pathway and other signalling pathways. This part has been completely rewritten but is still confusing because of some repetitions and inaccuracies, for example:

  • The same information is given in lines 27/28 than in lines 33/34: “inhibition of PI3K pathway activity enhanced the effects of SYK inhibition on AML cell viability and differentiation”!

  • The authors described the chemical inhibition of Syk on PI3K-Akt-mTOR pathway but never mentioned the chemical inhibition of the Ras-Raf-MEK-ERK pathway. In order to have homogeneous and complete information, the authors should refer to studies on MEK inhibitors and discuss the crosstalk pathway inhibition in AML.

page 14-15: Table 2

Instead of giving the potential, hypothetic advantages of combining these new drugs with PI3K-Akt-mTOR inhibitors, the authors should discuss published studies on drug combinations that have already been tested in AML.

The drug names are wrong and must be spelled correctly (Azacitidine, Venetoclax, Binimetinib, Midostaurin)!

page 16 : Title of table 2 is entirely false. It does not show studies with mTOR inhibitors but the authors listed new drugs in AML treatment and potential advantages to combine them with PI3K-mTOR-AKT inhibitors. The title must be changed in accordance with the contents of the table.

Minor points:

page 5 Figure 1 : the negative feedback loops must start from S6K and not from S6.

page 7 lines 33-34 : “mTORC1 activity is repressed by AMPK both represses indirectly, by activation of TSC2, and directly through phosphorylation of Raptor (38).” Authors have to keep the former version: “AMPK represses mTORC1 activity indirectly through phosphorylation and activation of TSC2, and directly through the phosphorylation of Raptor” for a better understanding of the message.

page 16 : the paragraph below table 2 legend must go!

page 6 lines 48-49 : “Phosphorylated S6K will sequentially phosphorylate IRS proteins, triggering their proteasomal degradation and inhibiting insulin/IGF-1-mediated PI3K activation.” References are missing.

There are a lot of typing errors and some grammatical mistakes, for example:

  • page 1 (line 20): “constitutive activation of the PI3K-Akt-mTOR pathway differs….”
  • page 3 (line 43): (IDH)1
  • page 4 (line 31) : “Phosphorylation of homologous residues in Akt2 and Akt3 occurs…”
  • page 6 (line 2):Both mTORC1 and PDK1 can directly activates S6K1, …”
  • page 6 (line 3): “ facilitates protein synthesis…”
  • page 6 (line 39): “PDK1 can directly phosphorylates S6K, …”
  • page 6 (line 46): “activating downstream mTOR Ssignalling”
  • page 8 (lines 11-12) : “Constitutive activation of this pathway has been observed in more than This pathway seems to be constitutively activated in 60% of AML patients, and this activation …”
  • page 8 (lines 30-31): “included 40 AML patients therein wich 21 patients (53%) patients demonstrated increased PI3K activation”
  • page 8 (line 37): “phosphorylation at Thr308 by PDK1 and at Ser473 by mTORC2 constitutively activates Akt..”
  • page 9 (line 5): “The Ras-Raf-MEK-ERK pathway strongly cooperates….”
  • page 9 (line 11): “Binding of an extracellular mitogen to a cell surface receptors activates the Ras-Raf-MEK-ERK pathway is a (100)…..”
  • page 9 (line 13): “Activated RS, by GTP binding…” RS?
  • page 10 (line 3): “and the PI3K-Akt-mTOR and
  • page 10 (line 4): “motility, and metabolism, and are activated…”
  • page 11 (line 1): “FOXO, BAD, GSK3) and PFK2.”
  • page 13 (line 21): “Given the limited effects of thisthese inhibitors so far in clinical…”
  • page 13 (line 36): “other mediators and siganlling signalling pathways…”
  • page 13 (line 38): “the effect of these inhibitors seems to variesvary…”
  • page 16: “The table summarize potential new drugs in AML therapy, and shortly addressing their potential benefits in combing combination with PI3K-mTOR-AKT inhibitors.”

Author Response

Reviewer 1

Major corrections have been made in the manuscript and recommendations have been taken into consideration by the authors.

However, a problem of clarity and synthesis persists in the parts of the text that have been rewritten. Despite adding a new table (table 2) listing new drugs used in AML treatment, the potential benefits of combining them with PI3K-Akt-mTOR inhibitors need to be completely re-discussed.

I recommend rewriting some parts of the manuscript to clarify the message.

Major points:

page 9: In paragraph 3.4, the authors described a crosstalk between PI3K-Akt-mTOR pathway and other signalling pathways. This part has been completely rewritten but is still confusing because of some repetitions and inaccuracies, for example:

  • The same information is given in lines 27/28 than in lines 33/34: “inhibition of PI3K pathway activity enhanced the effects of SYK inhibition on AML cell viability and differentiation

We are thankful for these comments. The reviewer is of course right that the PI3K-Akt-mTOR pathway has extensive crosstalk, some well described, and others more poorly understood. When it comes to the extent of an article like this, it is clear that all these interactions cannot be descried in detail, and we have to make compromises. However, we have again restructured this part in our manuscript, special avoiding unnecessary repetitions, and been more precisely regarding the interactions with the SYK and the Ras-Raf-MEK-ERK pathway.

  • The authors described the chemical inhibition of Syk on PI3K-Akt-mTOR pathway but never mentioned the chemical inhibition of the Ras-Raf-MEK-ERK pathway. In order to have homogeneous and complete information, the authors should refer to studies on MEK inhibitors and discuss the crosstalk pathway inhibition in AML.

We have included new further information regarding Ras-Raf-MEK-ERK pathway, special describing binimetinib, as an MEK inhibitor and. However, detail in this field is rapidly emerging, ad we guide to a new and updated reference regarding this topic (Digirmenci U, et al. Targeting Aberrant RAS/RAF/MEK/ERK Signaling for Cancer Therapy, Cells 2020; 13; 9(1).

page 14-15: Table 2

Instead of giving the potential, hypothetic advantages of combining these new drugs with PI3K-Akt-mTOR inhibitors, the authors should discuss published studies on drug combinations that have already been tested in AML.

The drug names are wrong and must be spelled correctly (AzacitidineVenetoclax, BinimetinibMidostaurin)!

page 16 : Title of table 2 is entirely false. It does not show studies with mTOR inhibitors but the authors listed new drugs in AML treatment and potential advantages to combine them with PI3K-mTOR-AKT inhibitors. The title must be changed in accordance with the contents of the table.

 We agree that this Table had some improvement points, and accordingly we have altered this table. Regarding clinical trial with these agents in combination with PI3K-mTOR-AKT inhibitors data are still lacking, and hence we believe this specific topic is a little premature for review yet. A more detail description of could let alone fulfill an own review article. In this setting we have tried to address all the criticism, at the same time try to focus on the main subject of this review, and guides to other articles for further reading.  

Minor points:

page 5 Figure 1 : the negative feedback loops must start from S6K and not from S6.

 Both figures are revised and made more quiescently regarding the use of arrows and lines in the figures.

page 7 lines 33-34 : “mTORC1 activity is repressed by AMPK both represses indirectly, by activation of TSC2, and directly through phosphorylation of Raptor (38).” Authors have to keep the former version: “AMPK represses mTORC1 activity indirectly through phosphorylation and activation of TSC2, and directly through the phosphorylation of Raptor” for a better understanding of the message.

 This is corrected

page 16 : the paragraph below table 2 legend must go!

As Table 2 is revised, this is also altered.

page 6 lines 48-49 : “Phosphorylated S6K will sequentially phosphorylate IRS proteins, triggering their proteasomal degradation and inhibiting insulin/IGF-1-mediated PI3K activation.” References are missing.

 This is added in our revised version.

There are a lot of typing errors and some grammatical mistakes, for example:

  • page 1 (line 20): “constitutive activation of the PI3K-Akt-mTOR pathway differs….”
  • page 3 (line 43):(IDH)1
  • page 4 (line 31) :“Phosphorylation of homologous residues in Akt2 and Akt3 occurs…”
  • page 6 (line 2):Both mTORC1 and PDK1 can directly activates S6K1, …”
  • page 6 (line 3):“ facilitates protein synthesis…”
  • page 6 (line 39):“PDK1 can directly phosphorylates S6K, …”
  • page 6 (line 46):“activating downstream mTOR Ssignalling”
  • page 8 (lines 11-12): “Constitutive activation of this pathway has been observed in more than This pathway seems to be constitutively activated in 60% of AML patients, and this activation …”
  • page 8 (lines 30-31):“included 40 AML patients therein wich 21 patients (53%) patients demonstrated increased PI3K activation”
  • page 8 (line 37): “phosphorylation at Thr308 by PDK1 and at Ser473 by mTORC2 constitutively activates Akt..”
  • page 9 (line 5): “The Ras-Raf-MEK-ERK pathway strongly cooperates….”
  • page 9 (line 11): “Binding of an extracellular mitogen to a cell surface receptors activates the Ras-Raf-MEK-ERK pathway is a (100)…..”
  • page 9 (line 13): “Activated RS, by GTP binding…”RS?
  • page 10 (line 3): “and the PI3K-Akt-mTOR and
  • page 10 (line 4): “motility, and metabolism, and are activated…”
  • page 11 (line 1): “FOXO, BAD, GSK3)and PFK2.”
  • page 13 (line 21): “Given the limited effects of thisthese inhibitors so far in clinical…”
  • page 13 (line 36): “other mediators and siganllingsignalling pathways…”
  • page 13 (line 38): “the effect of these inhibitors seems to variesvary…”
  • page 16: “The table summarize potential new drugs in AML therapy, and shortly addressing their potential benefits in combing combination with PI3K-mTOR-AKT inhibitors.”

We have proofread the manuscript, and errors are tried to be corrected.

Reviewer 3 Report

The additions have strengthened the review compared to its original form. There are still nomenclature, and consistency of naming Akt, or AKT, mTOR versus mTORC related issues

spelling of drug names should be verified. 

Author Response

The additions have strengthened the review compared to its original form. There are still nomenclature, and consistency of naming Akt, or AKT, mTOR versus mTORC related issues

spelling of drug names should be verified. 

We have revised our manuscript, and spelling is checked. The abbreviations are used consequently through the manuscript.

Round 3

Reviewer 1 Report

Major recommendations have been taken into account by the authors.

The message of the manuscript has been greatly improved and the description about combination therapy is more complete.

I would suggest the authors to watch out for some typing errors and some grammatical mistakes, for example:

  • page 8 (lines 7-8): “This pathway seems to be constitutively activated in 60% of AML patients, and this activation …”
  • page 9 (lines 2-3): “Binding of an extracellular mitogen to a cell surface receptors activates the Ras-Raf-MEK-ERK pathway is a (101)…”
  • page 9 (lines 6-7): “Activated RS, by GTP binding, simulate activates PI3K (103-105)…” + the authors should specify what “RS” means??? Ras?
  • page 9 (line 20): “This seems to be of special interest in AML cases…
  • page 10 (line 3): “and the PI3K-Akt-mTOR and
  • page 10 (line 4): “motility, and metabolism, and are activated…”
  • page 10 (line 18): “FOXO, BAD, GSK3) and PFK2.”
  • page 13 (line 21): “Given the limited effects of thisthese inhibitors so far in clinical…”
  • page 13 (line 27): “as the major new drugs that has have emerged as potential agents…”
  • page 14 (line 54): “the effect of these inhibitors seems to varies vary…”
  • page 15 (table 2) :
  • Potential synergism with potential to increase Akt suppression and by promoting expression of mTOR inhibitors such as PTEN.” Potential synergism through the increase of Akt suppression and the promoting of mTOR inhibitors expression such as PTEN.
  • “As SYK cross-reacts with the PI3K-Akt-mTOR, utilizing this potential it may be a more broadly applicable therapeutic strategy”.
  • “Potentiates the alterations in metabolism associated with PI3K-Akt-mTOR, especially in IDH1 mutated cases”
  • “Potentiates the alterations in metabolism associated with PI3K-Akt-mTOR, especially in IDH2 mutated cases”
  • page 17 (table 2 legend) : “benefits in combing combining them with…”
  • page 17 : the paragraph below table 2 legend must go!

Author Response

Again, we are grateful for the thoroughly comments from the receiver. We have utterly proofread our manuscript, and alterations in the text and improvement of grammatical mistakes are corrected. We agree that the comments from the reviewers indeed has helped us to improve our manuscript significantly.

We hope the manuscript could be consider for publication in International Journal of Molecular Science.

Reviewer 3 Report

I Like the improved and more comprehensive manuscript. I have no further comments.

Round 4

Reviewer 1 Report

The authors have shown a lot of efforts to improve the manuscript.

As a result, I now recommend the present form can be accepted for publication without further modification.